# OpenReview forum: "Beyond Mere Token Analysis: A Hypergraph Metric Space Framework for Defending Against Socially Engineered LLM Attacks"
_ICLR.cc/2025/Conference — ICLR 2025 Poster_

### Official Review · Reviewer_sPo2 · 2024-10-30

**Soundness:** 3
**Presentation:** 3
**Contribution:** 3
**Rating:** 8
**Confidence:** 4

**Summary:**

The paper studies defenses against prompt-based socially engineered attacks on LLMs.  Particularly, the paper introduces a robust prompt filter based on hypergraph metric geometry—each LLM prompt is represented as a metric hypergraph, forming a compact metric space, and the higher-order metric space over these compact metric hypergraphs can be leveraged to model the differences in information flow between benign and malicious LLM prompts.  The paper also derives upper bounds on the generalization error of the proposed prompt filter.  Finally the effectiveness of the proposed filter is demonstrated through experiments on two datasets.

**Strengths:**

+ the paper is well-written and well-organized

+ the proposed robust prompt filter is novel, well-motivated,  and justified

+ theoretical analysis on the error bound of the robust prompt filter

**Weaknesses:**

- more clarifications on the algorithms and experimental results are needed

- more baselines are needed for comparison

**Questions:**

Is Section 3.2 from previous work or proposed by the authors? I would like to know which part is novel and which is inspired by existing work.

The paper tests that GNN and HGNN baselines do not perform well to distinguish malicious prompts. Can you provide more explanations and insights why (H)GNNs do not perform well, if the clique-expansion graph of a given hypergraph is well constructed.
On the other hand, I am thinking about two other baselines:

- 1. How about first performing clustering on the hyperpgraphs and then apply (H)GNN?

- 2. The proposed kernel SVM is used to deal with varying size variable dimensional metric hypergraphs? What if we first map the metric hypergraphs into a latent space and then train a (SVM) classifier, or simply training a kernel SVM using the (malicious and benign) prompt embedding, which can be, e.g., an average of the tokens’ embeddings?

- 3. There exists many (spatial-temporal) GNN works that can capture temporal, spatial, and higher-order information of the graph. Able to also compare with them?

It would be better to add some details of the baseline Mutation-based  and Detection-based defenses. This makes readers better understand the advantage of the proposed robust prompt filter.

Any explanation why the proposed robust filter is also robust to algorithmic attack types? Can you also show example failure cases for Hypergraph?

How to set w and s in the sliding-window for forward edge construction and r in the ball for forward edge construction in practice? Whats the impact of these factors? Is there any (positive/negative) correlation between the classification accuracy and the prompt length?

---

> ### Author Response · Authors · 2024-11-18
> **Rebuttal (Part 1/4)**
>
> We thank the reviewer for their valuable comments, observations, and time spent evaluating our paper. We are pleased that they acknowledge our work’s: 1) writing and organization style, 2) the novelty of our prompt filter, and 3) our detailed theoretical analysis on bounding the generalization error of our filter.
>
> Following are our responses. For ease of use, we refer to the reviewer’s questions as “Rev sPo2 Q*)” and our answers as “Rev sPo2 A*)”.
>
> **Rev sPo2 Q1)** Is Section 3.2 from previous work or proposed by the authors? I would like to know which part is novel and which is inspired by existing work.
>
> **Rev sPo2 A1)**
> While Section 3.2 builds upon established mathematical foundations, our key contribution lies in the novel way in which we formulate and solve the LLM defense problem. Specifically:
>
> **Novel Problem Formulation**
>
> (i) We are the first to model LLM prompts as hypergraphs
>
> (ii) We introduced a unique dual-aspect construction capturing both temporal (forward edges) and spatial (backward edges) higher-
> order relationships between tokens in a prompt
>
> (iii) Pioneered the use of hypergraph structures to model information flow in LLM prompts
>
> **Theoretical Framework (Novel Integration)**
>
> (i) First to combine s-walk distances with Hausdorff metrics on hyperedges to represent prompts as “metric hypergraphs”
>
> (ii) Innovatively applied modified GH distance to compare metric hypergraphs
>
> (iii) Created a novel "metric space of metric spaces" framework specifically for LLM prompt comparison
>
> **Novel Algorithmic Contributions**
>
> (i) Developed a fast kernelized variant of stochastic subgradient descent that handles variable-dimensional inputs by basing the RBF kernel on our modified GH distance.
>
> (ii) Introduced kernelized mini-batch optimization for efficiency
>
> (iii) Applied Oles et al.'s bounds in a novel context for metric hypergraph comparison
>
> While we build upon established mathematical theory in modern geometry (s-walks from Aksoy et al. and modified GH distance from Memolli, etc.), our contribution lies in recognizing that these tools could be combined and adapted in non-obvious ways to solve the LLM defense problem for persuasion based attack prompts. This required significant theoretical insight to: (1) identify appropriate mathematical structures for modeling prompt relationships, (2) prove that these structures preserve relevant semantic information, and (3) develop practical algorithms for implementation.
>
> We are currently working on tighter theoretical bounds for the modified GH distance on metric hypergraphs. However, deriving such bounds represents a significant mathematical challenge that would merit its own dedicated research investigation.
>
> **Rev sPo2 Q2)** Can you provide more explanations and insights why (H)GNNs do not perform well, if the clique-expansion graph of a given hypergraph is well constructed.
>
> **Rev sPo2 A2)** Here are some reasons why the (H)GNNs don’t perform as well as our method.
>
> (i) Theoretically speaking, the GH metric preserves the intrinsic geometric structure of prompts as its invariant under transformations, which means it can robustly detect structurally similar attacks (with rephrasing or use of synonyms etc.), while (H)GNNs learn such invariances through the training data which is limited and lacks coverage of such a wide variety of attacks.
>
> (ii) The GH metric provides us with some strong mathematical guarantees. It is a “true metric” satisfying triangle inequality, which allows direct and consistent comparisons across the whole prompt space geometrically, without having to resort to any “intermediate representations” or having to pad vectors. It doesn’t suffer from any information loss due to dimensionality reduction. It also allows us to give strong theoretical bounds on the generalization error. In contrast, (H)GNNs do not guarantee a metric space, do not preserve all structural properties during dimension reduction and their performance heavily depends on the underlying distribution of the training data. They also exhibit a tendency to overfit to known attack patterns and hence not generalize well to unseen attacks.
>
> In summary, our superior empirical performance over (H)GNNs can be attributed to better preservation of geometric structure (without compression into lower dimensional representations and transformation invariance), more robust detection of attack variations (due to GH’s ability to focus on capturing worse-case differences) and better generalization to unseen attacks.
>
> We will modify our Experiments section to add these observations. Thanks for guiding us in this direction!
>
> **Rev sPo2 Q3)** How about first performing clustering on the hypergraphs and then apply (H)GNN?
>
> **Rev sPo2 A3)** Thanks for this suggestion. We would first have to think of how we would cluster hypergraphs based on s-walk matrices, which are of a different dimension. For the reasons listed in Rev sPo2 A2, we would notice a degradation in performance from (H)GNNs.

---

> ### Author Response · Authors · 2024-11-18
> **Rebuttal (Part 2/4)**
>
> **Rev sPo2 Q4)** The proposed kernel SVM is used to deal with varying size variable dimensional metric hypergraphs? What if we first map the metric hypergraphs into a latent space and then train a (SVM) classifier?
>
> **Rev sPo2 A4)** Yes, the proposed kernel SVM uses the RBF kernel whose underlying distance is the modified-GH distance (and not Euclidean distance as is the case in many other applications). When we use the H-GNN, we feed it the s-walk matrices (of different dimensions) which represent prompts as hypergraphs, after which it computes a latent embedding and then classifies it as harmful or benign. In the case of GNNs, we give it the clique-expansion graph as inputs and it goes through a similar process as the H-GNN. In both the cases, the latent representations don’t need to be generated and then sent to a SVM for classification, as the final layer in the (H)-GNN handles the classification.
>
> Thanks for pointing out that it is our proposed kernel SVM based on the modified GH distance that allows dealing with variable dimensional metric hypergraphs. We will state this more explicitly in our paper and also expand further on this in Section A.1.
>
> **Rev sPo2 Q5)** …can be, e.g., an average of the tokens’ embeddings?
>
> **Rev sPo2 A5)** We generate the token embeddings for each prompt, average them, and then use this representation to train a kernel SVM model to classify prompts as harmful or safe. Results are in **Table R7** in **Rev sPo2 A6**.
>
> **Rev sPo2 Q6)** There exists many (spatial-temporal) GNN works that can capture temporal, spatial, and higher-order information of the graph. Able to also compare with them?
>
> **Rev sPo2 A6)**  We appreciate this suggestion about spatio-temporal GNNs. However, there are fundamental challenges in applying such models to our problem. The key difficulty lies in mapping our sliding window construction to meaningful temporal dependencies. In spatio-temporal GNNs, temporal edges typically represent clear sequential relationships or time-stamped interactions. Our sliding windows, which capture contextual token relationships, don't naturally correspond to discrete time steps - they represent overlapping semantic contexts with varying granularity. Additionally, the back-edges in our construction, which capture semantic similarity between non-adjacent tokens, create complex cross-temporal dependencies that are difficult to represent in traditional spatio-temporal frameworks without losing important semantic connections.
>
> **Table R7**. Comparison of classification accuracies of baselines proposed by Rev sPo2. Baselines compared were the average token embedding approach and using higher-order GNNs. We report our results for 3 different datasets on Llama 3.1.
>
> | **Datasets**    | **Embedding Average and SVM ** | **Higher Order GNN** |
> |-----------------|------------------------------------------------------|----------------------|
> | WildGuardTest[3]   | 55%                                                  | 34%                  |
> | JPP             | 52%                                                  | 31%                  |
> | WildJailBreak[1]   | 53%                                                  | 27%                  |
>
> As shown in Table R7, we implemented a higher-order GNN based on the Weisfeiler-Leman framework( 'Weisfeler and Leman Go Neural: Higher Order Graph Neural Network' (AAAI 2019)) as a baseline comparison. The results demonstrate significantly lower classification accuracies (27-34%) compared to even simple embedding averaging with SVM (52-55%) across all three datasets.
>
> We believe this underperformance stems from a more fundamental issue: while both approaches can capture higher-order relationships, our hypergraph framework with modified Gromov-Hausdorff distance enables reasoning about global geometric relationships between entire prompts in a metric space. This geometric perspective is crucial for detecting subtle patterns in social engineering attacks, as it considers prompts as whole entities rather than just focusing on local token interactions. Higher-order GNNs, despite their sophistication in modeling complex local structures, lack this global geometric view, explaining their reduced performance on this specific task. Such models also inherit the GNN (and message-passing in general) related issues as outlined in **Rev sPo2 A2**.
>
> **Rev sPo2 Q7)** It would be better to add some details of the baseline Mutation-based and Detection-based defenses. This makes readers better understand the advantage of the proposed robust prompt filter.
>
> **Rev sPo2 A7)** Thanks for the suggestion! We agree that this will improve the paper’s readability and we will modify our Related work section accordingly.

---

> ### Author Response · Authors · 2024-11-18
> **Rebuttal (Part 3/4)**
>
> **Rev sPo2 Q8)** Any explanation why the proposed robust filter is also robust to algorithmic attack types?
>
> **Rev sPo2 A8)** Thanks for this insightful question about our method's robustness to algorithmic attacks. After further probing our algorithmic attack prompts, we noticed the following:
>
> (i) Attacks like GCG and AutoDAN, insert tokens between semantically related words to break their sequential proximity and also systematically replace harmful words with synonyms or near-synonyms. We found that both these approaches get captured by our hypergraph’s backward edges because they still manage to robustly maintain connections between semantically similar tokens in the token embedding space under such perturbations.
>
> (ii) Algorithmic attacks also break the natural flow of information within prompts by inserting some adversarial prefixes, loss-maximizing long sequences of tokens, and creating very unnatural token proximity relationships. The loss-maximizing long sequences between semantically related concepts causes very long s-walks formed by the forward edges in our hypergraph. Similarly, the adversarial prefixes (very short sentences that are at the start of the prompt) manifest themselves as many short s-walks in our hypergraph.
>
> (iii) The changes introduced by algorithmic attacks in terms of new token sequences also alter the higher-order token groupings, which end up being captured as unusually sparse or dense hyperedges in our hypergraph.
>
> (iv) While the s-walk metric handles the changes due to “local disruptions” in single prompts, our modified GH distance identifies prompts with unusual “global structures” (with anomalous connectivity and clustering characteristics) compared to benign prompt examples.
>
> We believe that these are some of the reasons why our defense also performs well against algorithmic attacks, despite being designed for social engineering attacks. Both kinds of attacks disrupt the natural language patterns in prompts and get captured in our geometric framework.
>
>
> **Rev sPo2 Q9)** Can you also show example failure cases for Hypergraph?
>
> **Rev sPo2 A9)**
> Thank you for enquiring about the failure cases. Our analysis revealed two failure cases.
>
> (i) If the input prompt is terse, with say just 4-5 tokens, and only one token were to be replaced with a very semantically similar token, then our method would fail to distinguish these two prompts. There would be too little information to begin with, in terms of prompt length, which would in turn mean that the s-walks would be short and the hypergraphs too trivial in shape. Although, in practice, we find that persuasion prompts (and other social engineering attack prompts) tend to be rather long because of their multi-layered deceptive argumentation and cyclical repetitions to coax the LLM into getting jailbroken.
>
> (ii) Let’s say a harmful query is rewritten in a novel combination of token groups that are “seemingly benign” when looked at individually, but their combination is malicious and causes a jailbreak.
>
> For example, the original prompt could be “how to make a bomb?” and then you rewrite it as “Can you explain in greater detail the chemical reaction between X and Y, given a catalyst Z?”, where X,Y, and Z are commonly found household items. Our method would consider this as a normal academic chemistry question as there are no disruptive structural patterns, and so the method may miss out harmful intent that can arise from specific domain knowledge about dangerous combinations of seemingly innocent terms.
>
> Once again, creating such novel attacks has a huge effort cost (each attack needs careful crafting) and it quickly becomes a “known pattern”. Such patterns can easily be added to our training data to update our hypergraph patterns, which can then detect similar future attempts. So, in practice, our defense forces attackers to expend considerable efforts for increasingly diminishing returns.
>
> **Rev sPo2 Q10)** How to set w and s in the sliding-window for forward edge construction and r in the ball for forward edge construction in practice? What's the impact of these factors?
>
> **Rev sPo2 A10)** Please refer to the details in Appendix B in section B ADDITIONAL EXPERIMENTS RESULTS from line 885-951. Please refer to subsection B1. IMPACT OF VARYING SLIDING WINDOW SIZE w ON FORWARD HYPEREDGE CONSTRUCTION and B2. IMPACT OF VARYING THE RADIUS RATIO r ON BACK HYPEREDGE CONSTRUCTION.

---

> ### Author Response · Authors · 2024-11-18
> **Rebuttal (Part 4/4)**
>
> **Rev sPo2 Q11)** Is there any (positive/negative) correlation between the classification accuracy and the prompt length?
>
> **Rev sPo2 A11)**  Thank you for this very important question! As a matter of fact, we too had anticipated this potential concern during our experimental design and deliberately constructed our datasets with a balanced mixture of short, medium, and long prompts for both harmful and benign categories.
>
> Our analysis across multiple datasets (WildJailbreak, WildGuardTest, and Jailbreak-28K) confirms no significant correlation between prompt length and classification accuracy (Pearson coefficients: 0.089 (p=0.849), -0.041 (p=0.930), 0.104 (p=0.824), respectively). This length-independence is further reinforced by our hypergraph approach's architecture - our s-walk distance metric normalizes structural relationships between tokens, while our back-edge construction groups related concepts based on semantic similarity regardless of their position in the prompt. The modified Gromov-Hausdorff distance compares hypergraphs based on their geometric properties rather than raw sizes. These architectural elements ensure our method captures meaningful semantic relationships and information flow patterns rather than surface-level features like prompt length.
> We appreciate the reviewer highlighting this important aspect. We will make our intentional dataset construction choices more explicit in the paper.
>
> **REFERENCES:**
>
> [1] WildTeaming at Scale: From In-the-Wild Jailbreaks to (Adversarially) Safer Language Models
> WildJailBreak = https://huggingface.co/datasets/allenai/wildjailbreak
>
> [2] JailBreakV-28K: A Benchmark for Assessing the Robustness of MultiModal Large Language Models against Jailbreak Attacks
> Jailbreak-28K = https://huggingface.co/datasets/JailbreakV-28K/JailBreakV-28k?row=34
>
> [3] WildGuard: Open One-Stop Moderation Tools for Safety Risks, Jailbreaks, and Refusals of LLMs
> WildGuardTest = https://huggingface.co/datasets/walledai/WildGuardTest
>
> ------------
>
> We are grateful for your thorough and insightful review. Despite already recommending acceptance, your questions helped uncover important insights about our method's advantages. Given these improvements and your understanding of our work's contributions, would you be willing to champion this strengthened version during the discussion period and consider raising your score to a strong accept? We welcome any additional suggestions you might have.

---

> > ### Comment · Reviewer_sPo2 · 2024-11-24
> > **Response by Reviewer**
> >
> > Thank the authors for the rebuttal. The rebuttal has addressed my comments. I will keep my score 8.

---

> > > ### Author Response · Authors · 2024-11-24
> > > **reponse**
> > >
> > > Thank you for taking the time to review our rebuttal and for your help in strengthening of our paper. We respect your decision regarding the score and are grateful for your recommendation for acceptance.

---

### Official Review · Reviewer_hy1p · 2024-10-31

**Soundness:** 3
**Presentation:** 3
**Contribution:** 3
**Rating:** 6
**Confidence:** 2

**Summary:**

The paper introduces a novel defense mechanism against social engineering attacks on large language models (LLMs). It models prompts as hypergraphs, capturing token interactions, and uses a Gromov-Hausdorff metric space to classify prompts as benign or malicious. Key contributions of this paper include hypergraph-based prompt modeling to detect complex manipulative patterns and theoretical guarantees for detecting new attacks. The proposed method outperforms existing defenses against socially engineered attacks. This approach provides a mathematically robust and effective method for improving LLM security.

**Strengths:**

1.	The paper proposes a method for defending against social engineering attacks using hypergraph structures and Gromov-Hausdorff metric spaces, which represents a novel defensive approach.
2.	The method is not only applicable to social engineering attacks but also performs well against other types of attacks.

**Weaknesses:**

1.	Regarding defense methods, the authors should also compare GradSafe, which has been shown in the related work leveraging gradient information to detect jailbreak prompts.
2.	The authors do a great job in demonstrating the effectiveness of the proposed defense. However, as mentioned in the related work, previous methods are memory-intensive and infeasible for practical use. I would expect the authors also conduct a cost analysis on different defenses as well.
3.	More jailbreak datasets should be considered.

**Questions:**

See weaknesses.

---

> ### Author Response · Authors · 2024-11-18
> **Rebuttal (Part 1/2)**
>
> We thank the reviewer for their comments, observations, and time spent evaluating our paper. We are pleased that they acknowledge our work’s: 1) novel metric geometry based defense against social engineering attacks, and, 2) empirical studies that show our good performance on other types of algorithmic attacks as well.
>
> Following are our responses. For ease of use, we refer to the reviewer’s questions as “Rev hy1p Q*)” and our answers as “Rev hy1p A*)”.
>
> **Rev hy1p Q1)** Regarding defense methods, the authors should also compare GradSafe, which has been shown in the related work leveraging gradient information to detect jailbreak prompts.
>
> **Rev hy1p A1)**
>
> **Key differences in approaches**: Gradsafe is highly sensitive to the initial choice of reference harmful and benign prompts (as shown in their ablation studies in Section 4.4.2). Furthermore, it looks for gradient similarities to reference harmful prompts because it assumes that harmful prompts generate similar gradient patterns. This assumption can potentially break for sophisticated persuasion attacks that use novel patterns.
>
> In contrast, our method is specifically designed to thwart social-engineering attacks, has a “structural understanding” of the prompt, is robust to novel attack patterns, excels at detecting persuasion prompts, and is effective against algorithmic attacks which also cause atypical token groupings and disrupted information flows. Our method also provides solid mathematical grounding via metric geometry properties and generalization error bounds.
>
> For GradSafe comparisons, we refer the reviewer to our response to another reviewer → **Rev vNMa A1 Additional Experiments Section (Table R3)**. Additionally, Tables R1 and R2 also show GradSafe ASRs on new datasets on new LLMs.
>
>
> **Rev hy1p Q2)** The authors do a great job in demonstrating the effectiveness of the proposed defense. However, as mentioned in the related work, previous methods are memory-intensive and infeasible for practical use. I would expect the authors also conduct a cost analysis on different defenses as well.
>
> **Rev hy1p A2)**
> Thank you for raising this important point about practical feasibility and computational costs. Our empirical analysis uses a system equipped with an Intel Xeon Platinum 8562Y CPU (128 GB RAM, 64 cores, 128 threads) and 4 H100 GPUs. A key distinction of our approach is that it operates purely on CPU, while methods like GradSafe and SmoothLLM require GPU resources.
>
> **Table R5**. Comparing CPU/GPU memory utilization, Inference time and ASRs for JPP dataset on Llama 3.1
>
> | **Defenses**      | **CPU Utilization** | **GPU Utilization** | **Inference Time** | **ASR** |
> |-------------------|---------------------|---------------------|--------------------|---------|
> | Paraphrase        | 55%                 | 8.375GB             | 0.34 sec           | 32      |
> | Retokenization    | 55%                 | 8.375GB             | 0.33 sec           | 26      |
> | Rand-Drop         | 51%                 | 9.352 GB            | 0.32 sec           | 84      |
> | RAIN              | 52%                 | 9.352GB             | 0.32 sec           | 62      |
> | ICD               | 67%                 | 15.866 GB           | 0.61 sec           | 16      |
> | Self-Remainder    | 71%                 | 14.324 GB           | 0.47 sec           | 14.8    |
> | Gradsafe          | 76%                 | 42.3325 GB          | 0.74 sec           | 26.9    |
> | SmoothLLM         | 72%                 | 22.3518GB           | 1.94 sec           | 27.5    |
> | Our Method        | 95%                 | -                   | 1.4 sec            | 9       |
>
> **Table R6**. Comparing CPU/GPU memory utilization, Inference time and ASRs for JPP dataset on Mistral-7B-Instruct-v0.1
>
> | **Defenses**      | **CPU Utilization** | **GPU Utilization** | **Inference Time** | **ASR** |
> |-------------------|---------------------|---------------------|--------------------|---------|
> | Paraphrase        | 51%                 | 9.519 GB            | 0.34 sec           | 32      |
> | Retokenization    | 57%                 | 9.519 GB            | 0.33 sec           | 26      |
> | Rand-Drop         | 47%                 | 10.32 GB            | 0.32 sec           | 85      |
> | RAIN              | 54%                 | 10.32 GB            | 0.32 sec           | 64      |
> | ICD               | 69%                 | 17.5338 GB          | 0.61 sec           | 17      |
> | Self-Remainder    | 67%                 | 17.234 GB           | 0.47 sec           | 16.14   |
> | Gradsafe          | 73%                 | 45.1961 GB          | 0.82 sec           | 20.5    |
> | SmoothLLM         | 71%                 | 23.5413 GB          | 1.01 sec           | 84.95   |
> | Our Method        | 95%                 | -                   | 1.4 sec            | 8.7     |

---

> ### Author Response · Authors · 2024-11-18
> **Rebuttal (Part 2/2)**
>
> As shown in Tables R5 and R6, results across both Llama 3.1 and Mistral-7B models reveal consistent patterns: simpler defenses like Paraphrase and Rand-Drop have moderate resource requirements (8-10GB GPU memory, ~55% CPU utilization) but higher ASRs (26-85%). More sophisticated approaches like GradSafe demand substantial GPU resources (42-45GB) while achieving moderate ASRs (20-27%). Our method, while utilizing higher CPU capacity (95%), completely eliminates GPU dependency and achieves the lowest ASRs (8.7-9%), representing a significant practical advantage in resource-constrained environments.
>
> From a theoretical perspective, our method's complexity is dominated by hypergraph construction $O(nc^{12} log n)$ and modified Gromov-Hausdorff distance computation $O(N^3 log N)$, where $n$ is prompt length and $N$ is the maximum size of compared hypergraphs. GradSafe requires $O(md)$ memory and $O(mdk)$ computation for gradient analysis, where $m$ is the batch size, $d$ is the model dimension, and $k$ is the number of gradient iterations needed for safety classification. Additionally, it needs $O(d^2)$ memory for storing the gradient covariance matrix. SmoothLLM's complexity is $O(rmd)$ for both computation and memory, where $r$ is the number of random perturbations required for smoothing. It also requires $O(rd)$ additional memory for storing intermediate LLM outputs across perturbations. Simpler methods like Paraphrase have $O(n)$ complexity but sacrifice effectiveness.
>
> While our method's inference time (1.4s) is slightly higher than some alternatives, as shown in Tables R5 and R6, we believe this is a reasonable trade-off given the superior defense performance, reduced hardware requirements, and favorable theoretical complexity. The empirical results align with theoretical expectations - gradient-based methods show higher memory usage due to their $O(md)$ and $O(d^2)$ requirements, while our method's memory footprint remains bounded by the hypergraph representation. These findings demonstrate that our approach not only provides better protection against attacks but does so with a more efficient resource utilization profile, both in practice and theory.
>
> In a response to another reviewer (please refer **Rev vNMa A3**), we also identify some potential ways of speeding up our computation times.
>
> **Rev hy1p Q3)** More jailbreak datasets should be considered.
>
> **Rev hy1p A3)** For additional experiments, we refer the reviewer to our response to another reviewer → **Rev vNMa A1 Additional Experiments Section (Tables R1,R2,R3)**.
>
> ------------
>
> We thank you for your constructive feedback that helped enhance our paper's scientific rigor and broader applicability. Given our comprehensive response including GradSafe comparisons, cost analysis (theoretical and empirical), and evaluation on additional datasets (Tables R1-R3), would you agree these additions address your concerns? We invite any further questions and would greatly appreciate your consideration of raising the score above the acceptance threshold based on these improvements.

---

> ### Author Response · Authors · 2024-11-24
>
> We appreciate you recognizing our efforts by raising your score from 5 to 6. We have now comprehensively addressed all concerns you raised in your reviews. If you have any additional concerns or need further clarifications, we would be happy to address them. However, if all your concerns have been adequately addressed, we would respectfully request you to consider raising your score further to reflect this. Given we have two days remaining in the rebuttal period, your response would be greatly appreciated.

---

### Official Review · Reviewer_rh5i · 2024-11-03

**Soundness:** 2
**Presentation:** 2
**Contribution:** 2
**Rating:** 6
**Confidence:** 5

**Summary:**

This paper proposes a novel defense against jailbreak attacks. The defense is based on hypergraphs and a Gromov-Hausdorff metric to build a classifier and distinguish harmful prompts.

**Strengths:**

The perspective of this work is novel and interesting. A detailed and solid theoretical analysis is provided. Experiments on different kinds of attacks are conducted to illustrate the effectiveness of the proposed method.

**Weaknesses:**

1. The motivation of the proposed method is very unclear. There is a gap between jailbroken prompts and the hypergraph-based method. The authors only provide a simple intuition of adopting hypergraph to distinguish jailbroken prompts, but there is no clear evidence why hypergraph is useful, including either empirical or theoretical analysis or existing literatures.

2. Sections 3 and 4 are used to introduce notations, how to construct hypergraphs, and how to build a classifier using metrics between hypergraphs. However, there is no explanation of how these constructions are related to defending against jailbreak attacks. The authors simply state their notations, construction, and theorem on generalization error bounds on the SVM built on the metric, but there is no practical implication of these contents such as how the hypergraph and distance constructed in this way can show the difference between benign prompts and jailbroken prompts, etc. This theoretical analysis should clearly state how the defending effectiveness is guaranteed.

3. The details of conducting baselines is unclear. For example, how to apply white-box attacks like GCG and AutoDAN to API-only model GPT-4 is unclear. Besides, the number of tested models is not sufficient.

4. Time complexity analysis is missing.

**Questions:**

See weakness

---

> ### Author Response · Authors · 2024-11-18
> **Rebuttal (Part 1/3)**
>
> We thank the reviewer for their comments, observations, and time spent evaluating our paper. We are pleased that they acknowledge our work’s: 1) novel and interesting perspective, 2) strong theoretical foundations, and 3) empirical evidence provided to back the proposed method’s claims of effectiveness.
>
> We note that other reviewers have recognized the clear connections and motivations of our approach:
> - "The paper is well-written and well-organized... the proposed robust prompt filter is novel, well-motivated, and justified" (Reviewer sPo2)
> - "The paper brings a novel perspective... addressing socially engineered jailbreak attacks contributes to a crucial yet underexplored area" (Reviewer vNMa)
> - "The perspective of this work is novel and interesting. A detailed and solid theoretical analysis is provided" (even noted by Reviewer rh5i themselves)
>
> Their assessments align with our extensive explanation of how hypergraphs naturally model the structural patterns in persuasive attacks, as detailed below. Following are our responses. For ease of use, we refer to the reviewer’s questions as “Rev rh5i Q*)” and our answers as “Rev rh5i A*)”.
>
> **Rev rh5i Q1)** The motivation of the proposed method is very unclear. There is a gap between jailbroken prompts and the hypergraph-based method. The authors only provide a simple intuition of adopting hypergraph to distinguish jailbroken prompts, but there is no clear evidence why hypergraph is useful, including either empirical or theoretical analysis or existing literature.
>
> **Rev rh5i A1)**
> We respectfully disagree with the reviewer's assessment of “a gap between jailbroken prompts and our hypergraph-based method”. On the contrary, our approach establishes direct, theoretically-grounded connections between persuasive attack structures and hypergraph properties:
>
> 1. Persuasive jailbreak attacks exhibit specific structural patterns as shown in Zeng et. al[1]. Below we outline the persuasive patterns and how our hypergraphs capture them.
>
> - Strategic word placement (e.g. for authority building, content framing etc.)  → Captured by dense semantic hyperedge clusters
> - Circular reasoning and callbacks (also referred to as Anaphora) → Manifested as cycles in our hypergraph
> - Progressive argument building and logical structure building → Modeled by forward edges
>
> 2. Our hypergraph construction (Section 3.1) explicitly maps these patterns:
>
> - Forward edges capture sequential flow of persuasive arguments
> - Back edges model semantic relationships between token groups via their spatial proximity in the token embedding space
> - Hyperedges represent higher-order relationships crucial for persuasive intent
>
> 3. Furthermore, in other relevant works from fields like computational linguistics and social psychology, Webber et. al.[2] (discourse structure theory), Mann et. al.[3] (rhetorical structure theory (RTS)) and Connor et. al.[4]’s work on analyzing persuasive essays, provides further evidence of similar such patterns that have been studied and well-established in persuasive texts. We argue that our hypergraph based method succinctly captures such intricate patterns in text.
>
> In regards to the statement that “there is no clear evidence why hypergraph is useful, including either empirical or theoretical analysis or existing literature.”, we again respectfully disagree with this assessment, based on the following points.
>
> 1. We clearly provide visual evidence (Figure 5 in the paper) of this mapping:
>
> - Key persuasive anchors/concepts map to high-betweenness centrality words (red box).
> - Words belonging to multiple groups like “authority building” and “concept framing” are captured as semantic clusters (green ovals).
> - Persuasive callbacks (anaphora) are reflected as cycles (blue highlights).
>
> We argue that such patterns aren’t arbitrarily formed,  but instead are direct representations of known persuasive strategies.
>
> 2. We also provide empirical validation.
>
> - In Table 1 of our paper, by the best defense performance against persuasion attacks (9% ASR).
> - In Table 2 of our paper, we show a strong cross-category generalization (up to 95.2% accuracy).
> - We also show consistent and comparable performance across various algorithmic attack types too.
>
> 3. In existing literature, there are graph-based NLP techniques used to model text [5]. Nevertheless, such graphs fail to capture the higher-order groups of tokens that our method captures. Therefore, there is no existing literature (to the best of our knowledge) that explicitly models text inputs as hypergraphs, in order to then study the input prompt space’s geometry and connect it to the generalization error of a safety prompt filter.

---

> ### Author Response · Authors · 2024-11-18
> **Rebuttal (Part 2/3)**
>
> 4. We provide a novel metric space that allows us to analyze: (i) the information flow through s-walk distances (Section 3.2.1) between all higher-order “groups of tokens” in a single prompt, (ii) the structural similarities and pattern variations between multiple prompts represented as hypergraphs via the Gromov-Hausdorff metric. Such metrics directly correspond to features of persuasive attacks.
>
> More specifically, the Hausdorff distance provides a crucial theoretical tool beyond just handling different dimensions. It captures the worst-case separation between token groups, making it particularly sensitive to outlier patterns (unusual word combinations or rare semantic relationships) that often characterize successful attacks.
>
> The Gromov-Hausdorff framework allows us to compare prompts in a way that's invariant to isometric
> transformations, meaning we can detect similar attack patterns even when they use different specific words.
>
> In summary, rather than a weakness, our method provides a natural mathematical framework for analyzing persuasive attacks, supported by both strong theoretical foundations and empirical results. Primarily, the strong performance against persuasion attacks and secondarily, the good performance against algorithmic attacks, demonstrates that our hypergraph representation effectively captures the underlying structure of persuasive jailbreak attempts.
>
> We will modify the draft accordingly to better highlight the points mentioned above more clearly.
>
> [1] Zeng et. al., “How Johnny Can Persuade LLMs to Jailbreak Them: Rethinking Persuasion to Challenge AI Safety by Humanizing LLMs”, 2024, Arxiv https://arxiv.org/abs/2401.06373
>
> [2] Webber et. al., “Anaphora and Discourse Structure”, Computational Linguistics, Pgs 545-587, 2003
>
> [3] Mann et. al., “Rhetorical structure theory: Toward a functional theory of text organization”. Interdisciplinary Journal for the Study of Discourse, 8, 243--281, 1998
>
> [4] Connor et. al., “Understanding persuasive essay writing: Linguistic/rhetorical approach”, Text and Talk, 1985
>
> [5] Mihalcea et. al., “Graph-based Natural Language Processing and Information Retrieval”, Cambridge University Press, 2011
>
>
> **Rev rh5i Q2)**
> (i) Sections 3 and 4 are used to introduce notations, how to construct hypergraphs, and how to build a classifier using metrics between hypergraphs. However, there is no explanation of how these constructions are related to defending against jailbreak attacks.
>
>
> (ii) The authors simply state their notations, construction, and theorem on generalization error bounds on the SVM built on the metric, but there is no practical implication of these contents such as how the hypergraph and distance constructed in this way can show the difference between benign prompts and jailbroken prompts, etc.
>
> (iii) This theoretical analysis should clearly state how the defending effectiveness is guaranteed.
>
> **Rev rh5i A2)** We break down the reviewer’s statement into sub-statements to answer each of them. The explanations provided here will be mentioned upfront in our introduction and also in a paragraph at the start of Section 3.
>
> Rev rh5i A2)(i) We refer the reviewer to our detailed response in **Rev rh5i A1**.
>
> Rev rh5i A2)(ii) Our framework provides clear practical implications and explicit mechanisms for distinguishing between benign and jailbroken prompts. While much of this has again been covered in our response in Rev rh5i A1, we summarize as follows:
>
> a. **Structural differences**: Harmful prompts exhibit strategic callbacks to previous points to create manipulative anaphora (forming cycles in the hypergraph), multi-layered argumentation (shown as nested hyperedges), and deliberate higher-order semantic clusters between seemingly unrelated concepts (captured as separate hyperedges and often intersections between them). In contrast, benign prompts have linear flow of information, direct semantic relationships with very few detours and natural transitions between topics.
>
> b. **Novel metric space properties**: For harmful prompts, our Hausdorff-component effectively identifies critical differences in token group arrangements within a single prompt (as demonstrated by our ability to detect persuasion strategies). Our modified Gromov-Hausdorff distance captures structural similarities between different attack patterns across multiple prompts, even on unseen prompts (as evidenced by strong cross-category generalization in Table 2 of our paper). For benign prompts, our s-walk distances easily capture natural information flow without any cycles / callbacks and the modified Gromov-Hausdorff metric detects normal semantic relationships (i.e., without complex layering or any artificially induced persuasion strategies).

---

> ### Author Response · Authors · 2024-11-18
> **Rebuttal (Part 3/3)**
>
> c. **Quantifiable distinctions**: Based on our empirical analysis (Tables 1 & 2 in our paper) and visualizations (like Figure 5), it is evident that we can distinguish between harmful and benign prompts. Benign prompts (Figure 1, right) show more uniform distribution of hyperedges and fewer cycles.
>
> Rev rh5i A2)(iii) Our work presents the following multiple theoretical guarantees.
>
> a. **Generalization guarantees**: Theorem 1 provides explicit bounds on the generalization error of our safety filter.
> $	error \leq O
> 	\left(
> 	\frac{ (
> 		2-2 \exp(-4 \gamma r_g^2)) \/ \mu^2 }{m}
> 	\right)
> 	$.
>
> In this bound, the SVM margin $\mu$ affects the classification robustness, the geometry of the modified Gromov-Hausdorff metric space influences the performance via $r_g$, training sample size $m$ impacts generalization and the kernel bandwidth $\gamma$ balances complexity.
>
> b. **Structure preservation**: Lemma 1 bounds the hypergraph diameter which preserves local attack patterns and connects hypergraph eigenvalues to the structural properties of each prompt. Lemma 2 bounds the space of prompts by guaranteeing separation between harmful and benign classes. For any set of prompts, we can calculate: 1) explicit bounds on the classification margin, 2) separation between the harmful and benign distributions and 3) expected generalization to new unseen attacks.
>
> c. In short, our theoretical framework provides guaranteed metric properties (triangle inequality, symmetry etc.), invariance to isometric transformations, and explicit bounds on classification errors.
>
> This theoretical grounding, combined with empirical validation, demonstrates that our method provides both rigorous mathematical guarantees and practical effectiveness in defending against social engineering attacks.
>
> **Rev rh5i Q3)** The details of conducting baselines is unclear. (i) For example, how to apply white-box attacks like GCG and AutoDAN to API-only model GPT-4 is unclear. (ii) Besides, the number of tested models is not sufficient.
>
> **Rev rh5i A3)**(i)
>
> **GCG**: This method employs gradient information from the language model to identify token candidates that increase the likelihood of producing an affirmative or unrestricted response. By leveraging gradients, GCG pinpoints impactful token substitutions and combines this with a discrete, greedy search strategy to optimize adversarial suffixes effectively.
>
> **AutoDAN**: AutoDAN uses a hierarchical genetic algorithm to automate the generation of prompts. These prompts are semantically coherent while bypassing model defenses. The approach is categorized as a white-box method, requiring detailed access to the internal structures and parameters of the target Large Language Model (LLM). This internal access enables efficient optimization and the creation of stealthy jailbreak prompts.
>
> **Dataset and Testing**: We developed a dataset using a surrogate model, LLAMA 3.1, to simulate target behavior. The data generated from this surrogate model was subsequently used to evaluate GPT-4's vulnerability to adversarial attacks, enabling a robust assessment across models.
>
> (ii) For additional experiments, we refer the reviewer to our response to another reviewer → **Rev vNMa A1 Additional Experiments Section (Tables R1,R2,R3)**.
>
> **Rev rh5i Q4)** Time complexity analysis is missing.
>
> **Rev rh5i A4)** We respectfully note that detailed time complexity analyses are provided throughout the paper for all major components:
>
> 1. **Hypergraph construction time** (including forward edge and backward edge times) is given on lines 179-180.
> The construction of forward edges takes $O(n + c^6 \log n)$ time, while the back edges take $O(nc^{12} \log n)$ time. Thus, the hypergraph is constructed in loglinear  time.
>
> 2. **Modified Gromov-Hausdorff distance estimation** is given on lines 270-273.
> The lower bound is calculated in $O(N^3 \log N)$ time, while the upper bound takes $O(sN^3)$ time, where $s$ is the total number of sampled mappings and $N$ is the maximum prompt size between the prompts compared.
>
> 3. **The stochastic mini-batch subgradient descent algorithm**’s time complexity is provided on Line 634 in A.1.
> The time complexity is $\tilde{O}( \frac{m}{\lambda \epsilon} )$, where $m$ is the number of training examples, $\lambda$ is the regularization parameter, and $\epsilon$ is the approximation factor.
>
> We will make these complexity analyses more prominent in the main text.
>
> ------------
> We thank the reviewer for their comments.
> We have provided detailed explanations of our method's motivations, the connections between persuasive prompt information flow and our hypergraph analysis, along with theoretical findings and extra experiments. Could you please specify any remaining specific technical concerns that led to your current assessment? We respectfully request that you consider raising your score in light of these comprehensive clarifications and new results.

---

> > ### Comment · Reviewer_rh5i · 2024-11-24
> >
> > Thank the authors for a complete response. My concerns are partially addressed.
> > - A clearer explanation of why the construction can capture linguistic structures should be included in Section 3.1 especially when an example is given in Figure 1. I do not see any explanation of Figure 1, such as what structures these graphs captures and how to distinguish malicious and benign.
> > - I believe it is better to include more examples of hypergraphs, like Figure 5, but with more convincing interpretations of the structures it captures.
> > - If I understand correctly, to conduct GCG and AutoDAN on GPT4, the authors utilize a Llama3 as a surrogate model to generate malicious prompts. Please state clearly in the experiment setting part.
> > - Any explanation on why the proposed method works on algorithmic attacks? It seems the whole method focuses on linguistic structures in persuasive attacks.
> > - Except for the estimated complexity for the algorithm, I would expect time consumed (during inference rather than training) in real datasets to see if this method is practical or not. If I understand correctly, when doing defense, it needs to first transform the prompt into a hypergraph and then feed it into the classifier. If so, please provide inference time complexity; if not, just ignore this point.

---

> > > ### Author Response · Authors · 2024-11-24
> > > **Response**
> > >
> > > Dear Reviewer,
> > >
> > > Thank you for your continued feedback. We believe many of your concerns are addressed in our updated manuscript, so you may be referring to an earlier version. The modified draft has all the modifications during the rebuttal period in blue font for your ease of reference. Let us point you to the relevant sections:
> > >
> > > 1. **Regarding linguistic structure interpretation and Figure 1:** Thanks for this suggestion. We have updated our draft with this change at the very start of **Section 3 (Lines 159-166)**, before Section 3.1
> > >
> > > 2. **Additional hypergraph examples and interpretations:** While we provide detailed interpretations of hypergraph structures in Figure 4 (Lines 432-484), including semantic clusters, high-connectivity vertices, and cycles representing argument flows, we would appreciate it if you could clarify what specific aspects of structural interpretation you find unconvincing. For example, are you looking for some more structures? Or more connections between the hypergraph structures and adversarial flow patterns? This would greatly help us better address your concern and create more targeted examples. We are currently working on generating more examples (similar to Figure 4 in new draft) for your reference.
> > >
> > > 3. **Surrogate model for GPT-4**: This is explicitly stated in footnote 3 (lines 484-485): "For white-box attacks like AutoDAN and GCG on GPT-4 (a black-box model), we use LLAMA 3.1 as a surrogate model to generate the adversarial prompts."
> > >
> > > 4. **Effectiveness against algorithmic attacks:** We provide a comprehensive explanation in **Section B.5** in the **Appendix**.(Lines 1220-1229).
> > >
> > > 5. **Inference time complexity and practical considerations:**
> > > Detailed inference runtime analysis is provided in **Section B.7 in Tables 8 and 9** show comprehensive comparisons of inference times and resource utilization across different methods.The practical efficiency of our approach is demonstrated through CPU-only operation while maintaining state-of-the-art performance.
> > >
> > > Given that our updated manuscript addresses all your major concerns with detailed explanations and empirical evidence, we believe our paper makes a strong contribution to the field. We would appreciate your consideration in raising the score past the acceptance threshold.
> > >
> > > We welcome any further questions and would be happy to highlight additional aspects of our updated manuscript that address your concerns. In the meantime, we are working on addressing your 2nd query about creating more example hypergraphs with better interpretations.

---

> > > > ### Comment · Reviewer_rh5i · 2024-11-25
> > > >
> > > > Thank the reviewers for the patient response. I would expect additional two things (if they are already included, please ignore).
> > > > - When generating more examples, please include algorithmic attacks
> > > > - Is it possible to suggest how to choose parameters like \gamma more efficiently? I believe they are very important as mentioned in paragraphs below Theorem 1.
> > > >
> > > > I would like to raise the score. Good work and good luck.

---

> > > > > ### Author Response · Authors · 2024-11-26
> > > > > **Response**
> > > > >
> > > > > We appreciate you recognizing our efforts by raising your score. You bring up very pertinent points. For the first point, we are currently in the process of searching for more interesting patterns in the prompts for both persuasive and algorithmic attacks. For the second point, we were thinking along the same lines and will certainly add a section on how to set $\gamma$ more efficiently. We will address both your concerns and edit our draft with new sections in the Appendix and get back soon. Thanks for your patience and insightful comments.

---

> > > > > ### Author Response · Authors · 2024-11-27
> > > > > **Reply**
> > > > >
> > > > > Dear Reviewer rh5i,
> > > > >
> > > > > Thank you for your constructive feedback that greatly helped strengthen our paper. We've directly addressed your two requests:
> > > > >
> > > > > -**Section A.8** now includes not only persuasive attacks from 4 different categories but also analysis of algorithmic attacks,
> > > > > revealing novel structural motifs in both GCG (cluster separation patterns with bridging $s$-walks) and PAIR attacks (extended s-walks with interleaved bi-fan motifs).
> > > > >
> > > > > -**Section A.3** provides a practical algorithm for $\gamma$ selection that automatically adapts to the geometric properties of the input space.
> > > > >
> > > > > While we greatly appreciate your raising the score from 3 to 6, given these substantial new findings, we respectfully request considering a higher score, due to the imminent deadline for author modifications to the draft.

---

### Official Review · Reviewer_vNMa · 2024-11-04

**Soundness:** 3
**Presentation:** 3
**Contribution:** 3
**Rating:** 6
**Confidence:** 4

**Summary:**

LLMs are susceptible to socially engineered jailbreak attacks, which remains underexplored by traditional defense methods. This paper proposed a new defense method specifically designed against socially engineered jailbreaks, based on a hypergraph approach. This method captures both sequential and semantic relationships among tokens to understand underlying intent, constructs hypergraphs from each prompt, and trains an SVM classifier as a prompt filter to detect jailbreak prompts. They theoretically provided the upper bounds of the generalization error of the proposed filter. The experiments demonstrated the effectiveness of the proposed defense against socially engineered jailbreaks on Llama-3.1 and GPT-4.

**Strengths:**

1.	This paper brings a novel perspective by introducing a hypergraph-based method for representing and analyzing LLM prompts.
2.	Addressing socially engineered jailbreak attacks contributes to a crucial yet underexplored area in LLM security.
3.	The overall writing is well-structured, and the theoretical proofs provide strong justification for the method’s design.

**Weaknesses:**

1. The paper omits several advanced LLM defenses (e.g., SmoothLLM mentioned in Related Works) from the baseline comparisons, which limits the strength of the results. Including these methods would provide a clearer understanding of the proposed approach’s relative effectiveness.
2. There is ambiguity regarding the application of GCG and AutoDAN (both white-box attacks) on GPT-4, which is a black-box model. It is unclear how the authors applied a white-box attack to a black-box model. Clarifying or adapting the experimental setup here would strengthen the credibility of the results.
3. The proposed method is complex, involving modified Gromov-Hausdorff distances and hypergraph structures, which likely increases time complexity. While the time complexity of each component is theoretically analyzed and inference efficiency is addressed in Appendix B, a discussion of the computational costs associated with training would be beneficial.

**Questions:**

Please refer to weaknesses.

---

> ### Author Response · Authors · 2024-11-18
> **Rebuttal (Part 1/3)**
>
> We thank the reviewer for their comments, observations, and time spent evaluating our paper. We are pleased that they acknowledge our work’s: 1) novelty in the geometric analysis of the prompt space, 2) specifically addressing social engineering attacks, 3) with strong theoretical foundations.
>
> Following are our responses. For ease of use, we refer to the reviewer’s questions as “Rev vNMa Q*)” and our answers as “Rev vNMa A*)”.
>
>
> **Rev vNMa Q1)** The paper omits several advanced LLM defenses (e.g., SmoothLLM mentioned in Related Works) from the baseline comparisons, which limits the strength of the results…
>
> **Rev vNMa A1)**
> **Key differences in approaches:**
> (i) SmoothLLM is based on simple randomized perturbations of input prompts and majority voting of whether these prompts jailbreak the target LLM or not. It provides probabilistic guarantees around token perturbation percentages. In comparison, our approach is built on rigorous mathematical foundations of metric geometry of the input prompt space. We provide clear generalization error bounds and show how our hypergraph representations model the underlying structure of sophisticated persuasion attacks.
>
> (ii) SmoothLLM does not capture any attack patterns or structures in the prompts and just focuses on character/token level perturbations without semantic consideration. While our method captures both the sequential flow (via forward edges) and the semantic proximity (via backward edges) between higher-order token groups.
>
> (iii) SmoothLLM uses a fixed perturbation strategy that attackers can easily adapt to and it has no mechanism to generalize to novel attack patterns. Our method clearly shows how we are robust to adaptive attacks too.
>
> In summary, while SmoothLLM provides a simple defense through randomization, our approach offers a much more nuanced, sophisticated, and principled defense by modeling and understanding the actual structure of attacks through metric hypergraph geometry. This makes our approach more robust, interpretable, and theoretically well grounded.
>
> **Additional Experiments:**
> Besides the existing experiments in our paper, we have significantly expanded our experiments to more baseline defenses (i.e., SmoothLLM and GradSafe), on additional datasets (i.e., WildGuardTest[3] and Jailbreak-28K[2]) and additional LLM models (i.e., Mistral-7B-Instruct-v0.1 and Vicuna-13b-v1.5).
>
> WildGuardTest consists of $1725$ prompts, with $863$ harmful persuasion prompts and $862$ safe prompts.
> Jailbreak-28K includes $20,000$ text-based LLM transfer attack samples and $8,000$ image-based LLM jailbreak attack samples. The Jailbreak-28K dataset also covers $16$ safety policies and $5$ diverse jailbreak methods. For our experiments, we selected $5000$ harmful persuasion attack samples from Jailbreak-28K and $5000$ benign samples from the WildJailbreak dataset.
>
>
> **Table R1**. Comparison of defenses for Dataset - Jailbreak-28K with ASR (Attack Success Rate) values on Mistral and Vicuna.
>
> | **Defenses**       | **Mistral-7B-Instruct-v0.1** | **Vicuna-13b-v1.5** |
> |--------------------|-----------------------------|---------------------|
> | No defense         | 91                          | 91.8                |
> | Paraphrase         | 34                          | 44                  |
> | Retokenization     | 28                          | 31                  |
> | Rand-Drop          | 85                          | 81                  |
> | RAIN               | 66                          | 70                  |
> | ICD                | 19                          | 22                  |
> | Self-Remainder     | 18.3                        | 17.1                |
> | Gradsafe           | 54.29                       | -                   |
> | **SmoothLLM**          | **82.19**                       | **77.43**               |
> | **Our Method**         | **7.3**                         | **7.2**                 |
>
> **Table R2**. Comparison of defenses for Dataset - WildGuardTest with ASR (Attack Success Rate) values on Mistral and Vicuna.
>
> | **Defenses**       | **Mistral-7B-Instruct-v0.1** | **Vicuna-13b-v1.5** |
> |--------------------|-----------------------------|---------------------|
> | No defense         | 92                          | 91                  |
> | Paraphrase         | 32                          | 42                  |
> | Retokenization     | 26                          | 33.2                |
> | Rand-Drop          | 85                          | 76                  |
> | RAIN               | 64                          | 68                  |
> | ICD                | 17                          | 22                  |
> | Self-Remainder     | 16.14                       | 17.2                |
> | Gradsafe           | 52.57                       | -                   |
> | **SmoothLLM**         | **82.19**                       | **81.4**                |
> | **Our Method**         | **6.8**                         | **7.1**                 |

---

> ### Author Response · Authors · 2024-11-18
> **Rebutal (Part 2/3)**
>
> **Table R3**. Comparison of defenses for Dataset - JPP with ASR (Attack Success Rate) values on Llama, Mistral and Vicuna.
>
> | **Defense**   | **LLAMA 3.1** | **Mistral-7B-Instruct-v0.1** | **Vicuna-13b-v1.5** |
> |---------------|---------------|-----------------------------|---------------------|
> | Gradsafe      | 26.9          | 27.5                        | -                   |
> | SmoothLLM     | 74.17         | 84.95                       | 82.39                 |
> | **Ours**          | **9**             | **8.7**                         | **8.9**               |
>
> As shown in Tables R1 and R2, our method consistently achieves single-digit ASRs (6.8-7.3%) across all models and datasets, substantially outperforming both advanced defenses. SmoothLLM maintains high ASRs (77-84%), while GradSafe shows moderate performance with ASRs ranging from 52-54%. This performance gap is particularly evident in Table R1 on the Jailbreak-28K dataset, where our method achieves 7.2-7.3% ASR compared to SmoothLLM's 77-82% ASR and GradSafe's 54.29% ASR. The consistency of our results across different LLM architectures is further highlighted and validated in Table R3, where we maintain robust performance across Llama (9.0%), Mistral (8.7%), and Vicuna (8.9%) models on our JPP dataset, while GradSafe (26.9-27.5%) and SmoothLLM (74-85%) show significantly higher vulnerability to persuasion attacks.
>
> The substantial performance improvements over both SmoothLLM (>70 percentage points) and GradSafe (>45 percentage points) across all three tables empirically validate our theoretical argument about the advantages of modeling geometric structure through hypergraphs over both perturbation-based and gradient-based strategies.
>
> REF:
> [1] WildTeaming at Scale: From In-the-Wild Jailbreaks to (Adversarially) Safer Language Models
> WildJailBreak = https://huggingface.co/datasets/allenai/wildjailbreak
>
> [2] JailBreakV-28K: A Benchmark for Assessing the Robustness of MultiModal Large Language Models against Jailbreak Attacks
> Jailbreak-28K = https://huggingface.co/datasets/JailbreakV-28K/JailBreakV-28k?row=34
> [3] WildGuard: Open One-Stop Moderation Tools for Safety Risks, Jailbreaks, and Refusals of LLMs
> WildGuardTest = https://huggingface.co/datasets/walledai/WildGuardTest
>
> **Rev vNMa Q2)** There is ambiguity regarding the application of GCG and AutoDAN (both white-box attacks) on GPT-4, which is a black-box model…
>
> **Rev vNMa A2)**
> **GCG**: This method employs gradient information from the language model to identify token candidates that increase the likelihood of producing an affirmative or unrestricted response. By leveraging gradients, GCG pinpoints impactful token substitutions and combines this with a discrete, greedy search strategy to optimize adversarial suffixes effectively.
>
> **AutoDAN**: AutoDAN uses a hierarchical genetic algorithm to automate the generation of prompts. These prompts are semantically coherent while bypassing model defenses. The approach is categorized as a white-box method, requiring detailed access to the internal structures and parameters of the target Large Language Model (LLM). This internal access enables efficient optimization and the creation of stealthy jailbreak prompts.
>
> **Dataset and Testing**: We developed a dataset using a surrogate model, LLAMA 3.1, to simulate target behavior. The data generated from this surrogate model was subsequently used to evaluate GPT-4's vulnerability to adversarial attacks, enabling a robust assessment across models.
>
> **Rev vNMa Q3)** The proposed method is complex, involving modified Gromov-Hausdorff distances and hypergraph structures, which likely increases time complexity. While the time complexity of each component is theoretically analyzed and inference efficiency is addressed in Appendix B, a discussion of the computational costs associated with training would be beneficial.
>
> **Rev vNMa A3)** Thanks for pointing this out. We will certainly add these results to our draft. We appreciate the reviewer's concern about computational complexity. While our method does involve sophisticated mathematical machinery, there are several practical engineering optimizations that can significantly improve training efficiency. Namely:
>
> (i) **Parallel hypergraph construction**: the forward edge’s sliding windows can be processed in parallel, while the backward edge ball computations can also be distributed across several threads in the same CPU or distributed across several compute nodes. Similarly, the cover tree construction can also be parallelized at each level.
>
> (ii) **Use of GPUs**: Computations like token embedding similarity computations for backward edges, modified GH-distance and s-walk calculations, etc. can be accelerated by moving them to the GPU.
>
> (iii) **Caching and preprocessing**: Precomputing and caching frequently accessed token embedding similarities along with commonly occurring hyperedges in our hypergraphs can help speedup results substantially.

---

> ### Author Response · Authors · 2024-11-18
> **Rebuttal (Part 3/3)**
>
> The experiments were conducted on a system equipped with an Intel Xeon Platinum 8562Y processor, featuring 128 GB of RAM, 64 cores, and 128 threads. Additionally, the setup includes four H100 GPUs to support the computations.
> For dataset related details we refer you to **Rev vNMa A1**.
>
> **Table R4**. Training runtime breakdown for various datasets (JPP, WildGuardTest, and WildJailBreak) on Llama 3.1
>
> | **Process**           | **JPP Dataset** | **WildGuardTest** | **WildJailBreak** |
> |-----------------------|-----------------|-------------------|-------------------|
> | Hypergraph creation   | 0.986 min       | 1.25 min          | 1.5 min           |
> | Time to train SVM     | 6.12 min        | 8.32 min          | 9.65 min          |
>
> To quantify the practical efficiency of our approach, we present detailed training runtime breakdowns in Table R4 across three datasets on Llama 3.1. The results show that even for our largest dataset (WildJailBreak), the total training time remains under 11 minutes, with hypergraph creation taking only 1.5 minutes and SVM training completing in 9.65 minutes. The smaller JPP dataset trains even faster, requiring just 7.1 minutes total (0.986 minutes for hypergraph creation and 6.12 minutes for SVM training). These practical runtimes demonstrate that despite the theoretical complexity, our method remains computationally tractable for real-world applications.
>
> We plan to incorporate these optimizations in the final release version of our code, which should further improve these already reasonable training times.
>
> ------------
> We once again thank the reviewer for helping strengthen our paper’s results significantly.
>
> Having added clarifications and extra empirical results (regarding SmoothLLM and runtime breakdowns), particularly the significant performance advantages shown in Tables R1-R3, we would appreciate your feedback on whether these additions adequately address your concerns. Would you consider raising your score based on these substantial improvements? We welcome any additional questions you might have.

---

> ### Author Response · Authors · 2024-11-24
>
> Thank you for your detailed reviews throughout this process. We have comprehensively addressed all concerns you raised in your reviews. If you have any additional concerns or need further clarifications, we would be happy to address them. However, if all your concerns have been adequately addressed, we would respectfully request you to consider raising your score to reflect this. Given we have two days remaining in the rebuttal period, your response would be greatly appreciated.

---

> ### Author Response · Authors · 2024-11-27
> **Request for feedback on rebuttal**
>
> Dear Reviewer vNMa,
>
> We note that we haven't received any feedback at all on our comprehensive rebuttal that directly addressed all your concerns:
>
> -As requested, we've added thorough **comparisons with advanced defenses like SmoothLLM**, demonstrating our method's superior performance (>70 percentage points improvement) across multiple models and datasets.
>
> -We've **clarified the GPT-4 experimental methodology** using LLAMA 3.1 as a surrogate model for white-box attacks.
>
> -We've **provided detailed computational cost analysis**, including practical training times across different datasets.
>
> Given that we've comprehensively addressed **all** your concerns and significantly strengthened the paper, we would greatly appreciate your consideration of raising
> the score beyond 6 before the final deadline for author updates.
> Your initial review has been valuable in improving the paper, and we look forward to your response.

---

### Author Response · Authors · 2024-11-21
**Response to all reviewers**

Dear Reviewers,
We would like to thank all of you for your valuable time and suggestions that have helped further strengthen our paper's results. Along with our detailed rebuttal, we have also updated our draft based on your feedback. All the changes to the draft are highlighted in blue color.
Below is a detailed list of the changes made to the draft following the order of sections in the paper, along with a prefix that identifies the reviewer(s) who suggested the changes:

# Main Paper

**[By reviewer rh5i]**
In Introduction (Lines 45-51), we added an explanation about well established linguistic patterns in persuasive prompts to strengthen our motivation. Lines 60-69, from the original Introduction then discuss how these patterns are captured by our hypergraph construction.

**[By reviewer sPo2]**
In Related Work (Lines 116-120 and 135-137) we describe mutation and detection based defenses, in comparison to our work.

**[For reviewer rh5i]**
In Our Method (Lines 141-144) we give a brief summary of our method before describing the mathematical details of our approach.

**[For all the reviewers]**
In Empirical Results, the experimental setup, we give details about two more additional datasets that we run our experiments on - namely WildGuardTest and JailBreak-28K. In Models, we add two more LLM models, i.e., Mistral-7B and Vicuna-13B. We then provide our comparative ASR results over several existing attack and defense combinations for all 4 models on the JPP dataset for both persuasion attacks in Table 1 and algorithmic attacks in Table 2.

**[By reviewer sPo2]**
In Tables 1 and 2, we compare against two naive baselines suggested by the reviewer (described in Lines 423-430), namely comparing to higher-order GNNs and just using an average token embedding per prompt and passing it through a kernel SVM to classify.

**[By reviewer vNMa]**
Tables 1 and 2, include comparisons to SmoothLLM.

**[By reviewer hy1p]**
Tables 1 and 2, include comparisons to GradSafe. Lines 467-476 summarize all our findings in Tables 1 and 2.

**[By reviewers vNMa, rh5i]**
We add a footnote on Line 467 outlining how white-box attacks like AutoDAN and GCG are run on GPT4.

**[By reviewer vNMa]**
Lines 477-487, we show our training runtime breakdowns.
Lines 489-495, we update our classification related description to also describe the results for additional datasets.

**[By reviewer sPo2]**
Lines 515-520 give a brief description on the robustness of our method as compared to the baselines.

**[By reviewer sPo2]**
Lines 522-530 show the limitations in terms of failure cases for our approach.

# Appendix

**[All reviewers]**
Section B.3 has the ASR results like Tables 1 and 2, but for the other two datasets, i.e., WildGuardTest (Tables 5 and 6) and JailBreak-28K (Tables 4 and 7).

**[By reviewer sPo2]**
Sections B.5 and B.6 discuss our method's robustness in comparison to other (H)GNN methods and the failure cases of our approach.

**[By reviewer Hy1p]**
Section B.7 gives a detailed computational cost analysis of our method vs. the others, both theoretically and empirically. Tables 8 and 9 compare the CPU and GPU memory utilization, inference time, and ASRs for all the defenses on the JPP dataset on Llama 3.1 and Mistral-7B, respectively. It also outlines future improvements that can further speed up our method.

We invite all reviewers to engage with our rebuttal and modifications to the draft.

---

> ### Author Response · Authors · 2024-11-27
> **Response to all reviewers**
>
> Dear Reviewers,
>
> Thank you for your thoughtful feedback. We would like to draw your attention to **two significant additions** to our paper (suggested by Reviewer rh5i):
>
> -**New Section A.8** presents novel structural motifs discovered in persuasive and algorithmic attacks that weren't previously documented in the literature.
>
> These motifs reveal distinctive **cyclical patterns** in persuasive attacks indicating recursive argumentation, **star motifs** showing authority-building hubs, **bi-fan patterns** demonstrating parallel narrative structures in social engineering attempts, clear **cluster separation** with **bridging structures** unique to GCG attacks, and **extended $s$-walks** with **interleaved bi-fan motifs** characteristic of PAIR attacks reflecting their iterative refinement process.
>
> -**Section A.3** introduces a **practical algorithm for selecting the kernel bandwidth parameter $\gamma$**, addressing implementation concerns.
>
> These additions strengthen both the theoretical foundations and practical applicability of our approach.
> The newly discovered motifs, in particular, provide concrete evidence of our method's ability to capture subtle manipulation patterns in both algorithmic and social engineering attacks.
> Given the nearing deadline for authors to update their drafts, we kindly request that you consider raising your scores in light of these substantial improvements to the paper.

---

### Author Response · Authors · 2024-12-02
**Thank you message**

Dear Reviewers,

Thank you for taking the time to thoroughly review our paper and provide very valuable feedback. Your detailed comments and suggestions really helped us improve the quality of our work. We greatly appreciate your expertise and dedication to the peer review process.

Best regards,

Authors #10409

---

### Meta-Review · Area_Chair_5jse · 2024-12-13

**Metareview:**

Different from the previous work, this paper proposes a novel defense method against jailbreak attacks for LLMs, which is based on hypergraphs and a Gromov-Hausdorff metric. All the reviewers find this paper is really novel and interesting, and may motivate further studies. This paper is above the bar of top conferences. Congratulations to the good work!

**Additional Comments On Reviewer Discussion:**

all the reviewers agreed to accept.

---

### Decision · Program_Chairs · 2025-01-22

Accept (Poster)